# Taxes, Inequality, and Equal Opportunities

**DOI:** 10.3390/e25091346

**Published:** 2023-09-16

**Authors:** José Roberto Iglesias, Ben-Hur Francisco Cardoso, Sebastián Gonçalves

**Affiliations:** 1Instituto de Física, Universidade Federal do Rio Grande do Sul, Porto Alegre 91501-970, RS, Brazil; sgonc@if.ufrgs.br; 2Instituto Nacional de Ciência e Tecnologia de Sistemas Complexos, CBPF, Rio de Janeiro 22290-180, RJ, Brazil; 3Escola de Gestão e Negócios, Programa de Pós-Graduação em Economia, UNISINOS, Porto Alegre 91330-002, RS, Brazil; 4Departamento de Economia e Relações Internacionais, Universidade Federal de Santa Catarina, Florianópolis 88015-400, SC, Brazil; benhur.phys@gmail.com

**Keywords:** econophysics, exchange models, inequality

## Abstract

Extreme inequality represents a grave challenge for impoverished individuals and poses a threat to economic growth and stability. Despite the fulfillment of affirmative action measures aimed at promoting equal opportunities, they often prove inadequate in effectively reducing inequality. Mathematical models and simulations have demonstrated that even when equal opportunities are present, wealth tends to concentrate in the hands of a privileged few, leaving the majority of the population in dire poverty. This phenomenon, known as condensation, has been shown to be an inevitable outcome in economic models that rely on fair exchange. In light of the escalating levels of inequality in the 21st century and the significant state intervention necessitated by the recent COVID-19 pandemic, an increasing number of scholars are abandoning neo-liberal ideologies. Instead, they propose a more robust role for the state in the economy, utilizing mechanisms such as taxation, regulation, and universal allocations. This paper begins with the assumption that state intervention is essential to effectively reduce inequality and to revitalize the economy. Subsequently, it conducts a comparative analysis of various taxation and redistribution mechanisms, with a particular emphasis on their impact on inequality indices, including the Gini coefficient. Specifically, it compares the effects of fortune and consumption-based taxation, as well as universal redistribution mechanisms or targeted redistribution mechanisms aimed at assisting the most economically disadvantaged individuals. The results suggest that fortune taxation are more effective than consumption-based taxation to reduce inequality.

## 1. Introduction

In capitalist economies, social and economic inequality has become an ingrained characteristic. While a certain degree of inequality can serve as a motivator for individuals to strive for progress, excessive inequality poses a significant barrier to the fundamental driver of the economy: trade. Consequently, individuals in dire poverty are marginalized from participating in economic transactions, resulting in reduced circulation of money and diminished consumption of goods. From the insights of Adam Smith [1] to the perspectives of contemporary neo-liberals [2], including proponents of the Austrian School and minimal state intervention [3], orthodox economic theory has long posited that the inherent mechanisms of the market will naturally alleviate disparities in wealth. This theory argues that by providing individuals with opportunities for advancement, everyone will have a chance to improve their circumstances, thereby reducing inequality [4]. One of the early studies in income distribution was developed by Italian economist Vilfredo Pareto [5].

Pareto’s analysis of income data from the 19th century revealed a striking phenomenon: income distribution follows a power law, with the exponent now known as the Pareto exponent. He went beyond this observation, asserting that the non-Gaussian nature of income distribution suggested that individuals or enterprises took deliberate actions leading to higher income. This notion still finds support today, despite the fact that there are more critics of meritocracy [6,7] than proponents. In any case, a persistent hypothesis prevails in capitalist societies: that with “equal opportunities”, individuals of sufficient intelligence and effort can ascend the social pyramid.

Nevertheless, the current state of affairs deviates markedly from this idyllic conception. Recent data from the USA paint a highly contrasting picture: a mere 1% of the economic elite possesses almost half (50%) of the total wealth, with the top quantile (20%) of the population owns a overwhelming 88% of available resources, as indicated by Wolff’s research in 2017 [8]. As highlighted by Piketty [7], the chronological progression of the wealthiest segment of society’s assets expands at a swifter pace than the overall economy. Adding to this stark reality, the once-promising notion of social mobility stands exposed as a mere illusion, as considerable fortunes persist through the channels of inheritance, as highlighted in Fernholz’s work in 2023 [9].

Even simple exchange models used in econophysics to simulate trade and economic exchanges demonstrate this phenomenon. In two recent papers [10,11], we have demonstrated that exchange models considered *fair*, where agents participating in trade have equal chances of earning money, inevitably lead to the total concentration of wealth in the hands of a single individual or a select few. Moreover, most microscopic models of exchange among economic agents exhibit this behavior (see ref. [12]).

These models consider an ensemble of interacting agents that exchange a fixed or random amount of their total wealth. The exchanged wealth is susceptible to several interpretations. It could be the money given for some service or commodity or an *error* during the exchange [13], and it may be attributed to a profit or *plus valia*. Analogous to physical systems where particles exchange energy through binary conservative collisions, these models [14,15,16,17] consider a set of interacting agents that exchange wealth. If the exchanged amount of wealth is a random fraction of the wealth of each agent, the resulting wealth distribution follows a Gibbs exponential distribution [17]. However, such models lack fairness, as the values each agent puts at stake may differ significantly.

One of the most used models of wealth exchange among economic agents is the so-called Yard-Sale model. This model, in its original version, is a fair model because each agent has the same possibility of winning the same amount of money. The basic idea is that no one can receive, in any trade, more than he/she is putting at stake during the exchange.

Numerical [12] and analytical [18] results with the Yard-Sale model, or some variations of it, point to *condensation*, i.e., a continuous concentration of all available wealth in just one or a few agents, leading to an absorbing state where no more wealth is exchanged [10]. The phenomenon of *condensation*, while well-known to experts in the field, might appear to challenge a fundamental principle of thermodynamics because it leads to a situation seemingly at odds with the second law of thermodynamics. In the conventional Kinetic Theory of Gases, as formulated by Boltzmann, random energy exchanges propel the system towards the equal distribution of energy, culminating in a state of maximum entropy. However, when energy (or wealth) exchanges are restricted from exceeding the inherent energy, a distinct scenario unfolds. This outcome corresponds to a state of minimum entropy or, conversely, maximum information. While the second law predicts a *thermal death of the universe*, characterized by the uniform distribution of energy and a uniform temperature, alternative models of equitable exchange envision a *thermal death of trade*, marked by large disparities in wealth (comparable to temperature differences), ultimately leading to a cessation of trade. Nevertheless, the current path of the global economy, characterized by the persistent growth of inequality [7], seems to bring us uncomfortably close to this ominous scenario.

Different modifications have been introduced in the Yard-Sale model to overcome condensation. For example, increasing the probability of favoring the poorest agent in a transaction [19,20] or introducing a taxation mechanism [13,21,22], wherein all agents periodically contribute taxes, and the collected amount is subsequently distributed among them. This approach closely resembles real-world political systems adopted by various countries. Therefore, our focus will be on examining the impact of taxes on wealth redistribution and inequality reduction.

In the following section, we will describe the exchange model we are going to use, i.e., the Yard-Sale model. In Section 3, we review previous findings with taxation on wealth; in Section 4, we present the novel results with taxation on exchanges and redistribution, and the impact in reducing inequality.

## 2. The Model

We consider an ensemble of interacting economic agents, where two of them are selected sequentially and at random to exchange a predetermined fraction of their wealth. Agents do not risk all of their capital in each exchange, but they save a fraction, which depends on their risk aversion [19,20,23,24,25,26]. Therefore, the attributes of each agent *i* are the risk-aversion factor $\beta_{i}$ and its wealth $w_{i}$. Both are initially drawn from random uniform distributions in the [0,1) interval (for the exchange-tax system, we use another distribution for β, that is the same value for all agents, therefore becoming a parameter of the model, which is varied to check on its effect), but while $\beta_{i}$ stays constant for each agent, $w_{i}$ (simulations are insensitive to the initial distribution of $w_{i}$) changes because of exchanges involving that agent.

It is worth noting that certain models introduce the possibility of wealth creation or destruction during these exchanges [27]. However, for the purposes of our discussion, we will limit ourselves to conservative models, where the total wealth remains constant.

Let us assume an exchange of wealth between agents *i* and *j*. Supposing that *i* wins an amount of wealth from *j*; we have that
$$w_{i}^{*}=w_{i}+dw\;\;\;\mathrm{and}\;\;\;w_{j}^{*}=w_{j}-dw,$$
where $w_{i(j)}^{*}$ is the wealth of the agent i(j) after the exchange.

The most widely used rule to determine the quantity dw transferred from the loser to the winner is the *fair* one, which states that $dw=\min[\left(1-\beta_{i}\right)w_{i}\left(t\right);\left(1-\beta_{j}\right)w_{j}\left(t\right)]$. It is considered *fair* because the amount of wealth exchanged is the minimum of the quantities risked by the two agents and the same regardless of who wins, and it is the basis of the Yard-Sale model [28].

As we stressed before, numerical and analytical studies with the Yard-Sale model, as well as its variations, consistently lead to condensation. Recently, we have given a general proof that all models following a *fair* principle, including the Yard-Sale, inevitably lead to condensation [10,11]. To overcome this fate, different rules of interaction have been applied, for example increasing the probability of favoring the poorer agent in a transaction [19,20] or introducing a cut-off that avoids interactions between agents below and above this cut-off [29]. One particular choice is to use a rule suggested by Scafetta [12,19], where, in the exchange between the agents *i* and *j*, the probability of favoring the poorer partner is given by the following:(1)$$p=\frac{1}{2}+f\times \frac{|w_{i}\left(t\right)-w_{j}\left(t\right)|}{w_{i}\left(t\right)+w_{j}\left(t\right)},$$
and *f* is a factor that we call the *social protection* factor, which goes from 0 (equal probability for both agents) to 1/2 (highest probability of favoring the poorer agent). In each interaction, the poorer agent has a probability *p* of earning a quantity dw, whereas the richer one has a probability of 1−p. It is evident that the higher the difference in wealth in a given pair of agents, the higher the influence of *f* in the probability; thus, *f* is a good indicator of the degree of application of social policies of wealth distribution. This rule have been studied in full generality in some previous articles [12,30].

We have provided a concise overview of the impacts of the social protection factor. For a more in-depth examination of this approach to diminishing inequality, we direct interested readers to the comprehensive review by Chakraborti et al. [31,32].

While this simple mechanism helps to reduce inequality, some critics argue that real-world exchanges tend to favor wealthier agents. In addition, a consensus has not yet been reached on how to accurately correlate the protection factor with tangible economic measures. Consequently, it seems that a more logical way to reduce inequality is through the redistribution of the taxes collected. Therefore, we will focus on the effects of taxes. In the next section, we will delve into the simplest tax system: tax on wealth.

## 3. Taxes on Fortune

In this section, we present previously published results [33] concerning the implementation of a simple flat tax on wealth. Our simulation revolves around a society consisting of N agents who engage in wealth exchanges based on the Yard-Sale model. At each time-step, two agents are randomly selected, facilitating a monetary exchange where one participant emerges as the winner while the other becomes the loser. Regarding the tax collection mechanism in our simulation, it operates as follows: after every Monte Carlo Step (MCS), i.e., following N/2 exchanges, all agents contribute the same fraction λ of their wealth as taxes. (It is worth noting that the wealth tax shares similarities with property or fortune taxes, albeit being less prevalent than income taxes.) Consequently, the redistribution of money can manifest in two distinct ways: a universal allocation, wherein funds are distributed evenly among the entire population, or a focused approach, wherein the funds are specifically directed towards individuals with lower wealth.

All results presented here are averages over $10^{3}$ samples for three system sizes *N*: $10^{3},10^{4}$, and $10^{5}$. As the obtained results are almost independent of the size, we have plotted just the outcome for *N* = $10^{5}$ and *N* = $10^{4}$ agents. The saving propensity factor β, as well as the initial wealth of each agent, are chosen at random from a uniform probability distribution in the interval (0,1). While the individual wealth changes along the simulation because on the exchange interactions, the saving factor of each agents is fixed.

### 3.1. Universal Redistribution

The most straightforward type of redistribution is universal, wherein the entire tax revenue collected is distributed equally among all individuals, irrespective of their wealth. Similar taxation mechanisms have been proposed in prior studies [13,21], albeit with the assumption of β values close to 1 and in the context of the small transaction limit approximation. Notwithstanding these differences, our findings, which have been published elsewhere [33], qualitatively correspond with prior research; they show that the Gini index decreases as the tax percentage increases, as expected; so, the taxation mechanism can effectively mitigate inequality. However, the effect of the tax percentage is non monotonic; indeed, it is more effective at small values. Effectively, 10% of taxes makes a huge change in the Gini index, lowering it from 1 (no taxes) to 0.5, while increasing taxes up to 25% lowers the Gini index to 0.3. More details on the quantitative effect of the universal taxes on the Gini index can be seen in Figure 1-right, (blue curve). We cannot expect, in real societies, a tax percentage above 25%. Recent contributions [22,34] have explored related systems of universal redistribution, where the tax percentage depends on wealth.

### 3.2. Focused Redistribution

In the targeted scenario, the total tax collection is distributed among the *p* poorest fraction of the population, referred to as the targeted population. The universal case corresponds to p=1. Figure 1-left illustrates the relationship between the Gini index and both λ and *p*. Notably, when the allocation is limited to less than 1% of the population ($p\leq 10^{-2}$), which aligns with many governmental initiatives aimed at assisting the unemployed and extremely impoverished individuals, the impact on the Gini coefficient is almost negligible. To achieve a noticeable effect in reducing inequality, it becomes necessary to extend assistance to at least the poorest 3–4% of the population. Additionally, Figure 1 reveals an optimal value of $p=p^{*}$ that minimizes inequality for each tax rate λ, indicating an intriguing non-trivial relationship between λ and *p* in this context.

Finally, in Figure 1-right, we compare the Gini index as a function of λ for two cases: p=1 (universal case) and $p=p^{*}$ (optimal targeted case). Notably, for intermediate values of λ, particularly around λ≈0.3, the regulatory mechanism of assisting only a fraction of the population proves to be more effective in significantly reducing inequality.

## 4. Taxes on Exchanges

One common taxation in many countries is the VAT (value-added tax), or IVA in Spanish-speaking countries. A slightly different tax, called ICMS (tax on the circulation of goods and services) is applied in Brazil. This is a tax that everybody pays when buying goods or paying for services.

We simulate the VAT system by taxing each exchange with a fixed percentage on the exchanged quantity dw. In practice, the tax collection works as follows: two agents, *i* and *j*, are randomly selected to exchange wealth in such a way that agent *j* will lose an amount of wealth $\left(1-\beta \right)\min\left(w_{i},w_{j}\right)$ while agent *i* will receive this value reduced by a factor (1−λ). Thus,
(2)$$w_{i}^{*}=w_{i}+\left(1-\lambda \right)\left(1-\beta \right)\min\left(w_{i},w_{j}\right)\;\;\;\mathrm{and}\;\;\;w_{j}^{*}=w_{j}-\left(1-\beta \right)\min\left(w_{i},w_{j}\right),$$
where λ is the tax rate. The collected taxes $\lambda \left(1-\beta \right)\min\left(w_{i},w_{j}\right)$ of each exchange are accumulated during one MCS, that is, along N/2 exchanges. After this period, the collected taxes are equally distributed among all agents. We denote the liquidity of the system *L* as the total value received by the agents in exchanges along one MCS. The reader may have already noticed that here, we make use of a constant and universal saving factor β, in order to simplify the calculations. But, there are no obstacles to using an individual $\beta_{i}$ for each agent.

As before, we again use the Gini index to measure inequality. We show in Figure 2 the Gini index as a function of the tax fraction λ for different values of β. We observe that the higher the tax rate, the lower the inequality, as expected, and inequality also decreases if the risk aversion increases, similarly to what was obtained in the models without taxes. In the trivial case (λ=0), we recover the G=1 result (condensation).

In Figure 3, we depict the liquidity as a function of λ, for different values fo β. Here, an interesting feature is observed. While the behavior of liquidity with β is not simple, it generally decreases as the risk aversion increases, which is expected. However, for each value of β, there is an optimum value of $\lambda =\lambda^{*}$, such that the liquidity $L=L^{*}$ is maximum—and inequality is minimum. It is clear that very low or very high taxes are a burden to trade; therefore, an intermediary, not trivial value appears as a function of β to maximize liquidity. Nevertheless, such maximization has to be counterbalanced with the minimization of the Gini index. In the next figure, Figure 4, we show that the optimum tax rate ($\lambda^{*}$) decreases as a function of β.

Finally, in Figure 5, we show how the Gini index and liquidity behave as a function of β, when the optimum tax rate is applied. We can observe a a trade-off between equality and liquidity for different values of β.

## 5. Discussion and Conclusions

Recent studies in the field of econophysics have unveiled an intriguing phenomenon: fair models that allocate equal chances of winning to individuals may, in fact, result in maximum inequality. This implies that despite initially equal opportunities, there is a need for redistribution mechanisms to ensure greater equality in outcomes.

In this article, we delve into the topic of taxation and explore how different taxation mechanisms can contribute to reducing inequality. Specifically, we juxtapose the findings of earlier research [10]—which delved into wealth taxes—with the concept of transaction taxes—akin to a value-added tax (VAT) on consumption. Taxation can be a potent tool for redistributing wealth and resources from the affluent to the less privileged. However, the type of taxation system implemented can have vastly different impacts on the level of inequality in society.

Through our research, we analyze various types of taxation models, including a flat wealth tax with universal and directed redistribution, as well as a wealth-transaction tax with universal redistribution.

Our research has identified that a small fraction of wealth tax can significantly contribute to reducing inequality. Our analysis reveals that by implementing targeted redistribution mechanisms that specifically cater to the poorest individuals in society, the impact of wealth tax can be even stronger. By providing resources and support to those who are most in need, we can foster greater social and economic equality.

However, while a higher tax rate consistently leads to a decrease in inequality, the volume of economic activity follows an inverted U-shaped curve in response to changes in the tax rate. In other words, we can identify an optimal tax rate that maximizes economic activity. Nevertheless, the specific optimal tax rate varies depending on the average saving rate of individuals in a society. We have obtained an optimal tax rate that can range from 0.25 to 0.55, depending on the average saving rate.

When analyzing the outcomes, it is crucial to acknowledge that we are working within a basic fair exchange model. Despite retaining the fundamental aspects of trade, this model does not encompass goods production or economic expansion. These constraints mean that while the impact of taxes and redistribution evidently diminishes inequality, as seen in the practices of certain nations through social allocations, the numerical outcomes should be considered as instructive rather than definitive predictions.

To sum up, even though equitable models that distribute equal opportunities might seem just, they can paradoxically lead to heightened disparities in actual outcomes. This research underscores the importance of adopting efficient redistribution mechanisms, such as levying wealth taxes on both wealth and transactions—where the former proves more effective than the latter—in order to foster heightened levels of societal and economic parity.

## Figures and Tables

**Figure 1 entropy-25-01346-f001:**
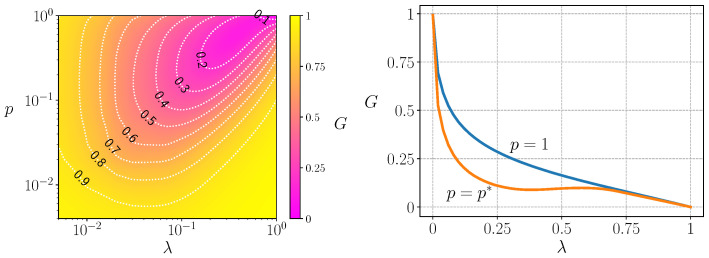
(**Left**): Gini index represented as a heat map plus iso-Gini curves, as a function of the tax fraction (λ) and the bottom fraction of agents that receive the collected taxes (*p*). (**Right**): Gini index as a function of λ for p=1 (universal case) and $p=p^{*}$ (optimal targeted case). Both figures were obtained for a system of $N=10^{4}$ agents (from ref. [33]).

**Figure 2 entropy-25-01346-f002:**
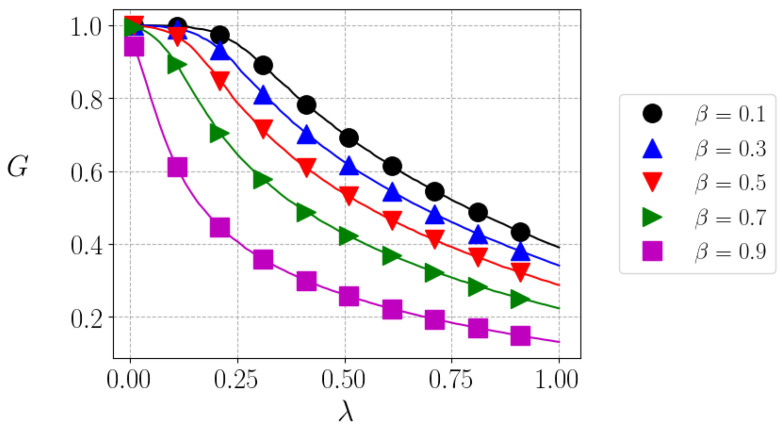
Equilibrium Gini index as a function of λ, the tax index, for different values of β. Lines correspond to simulations with $N=10^{4}$ agents and symbols, with $N=10^{5}$.

**Figure 3 entropy-25-01346-f003:**
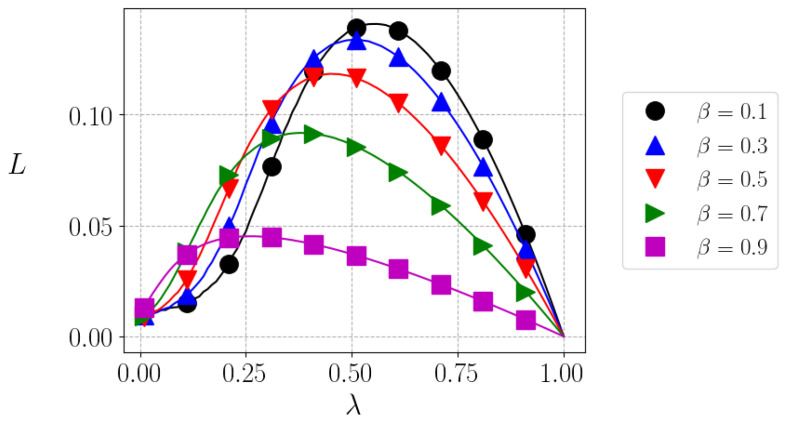
Equilibrium liquidity as a function of λ, the tax index, for different values of β. Lines correspond to simulations with $N=10^{4}$ agents and symbols, with $N=10^{5}$.

**Figure 4 entropy-25-01346-f004:**
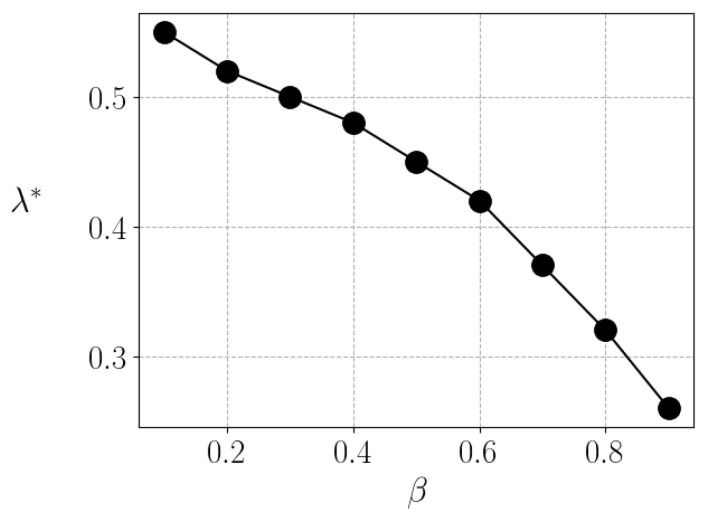
Optimum value of λ as a function of d β. The line is just a guide for the eyes, points are the results of the simulations for different values of β.

**Figure 5 entropy-25-01346-f005:**
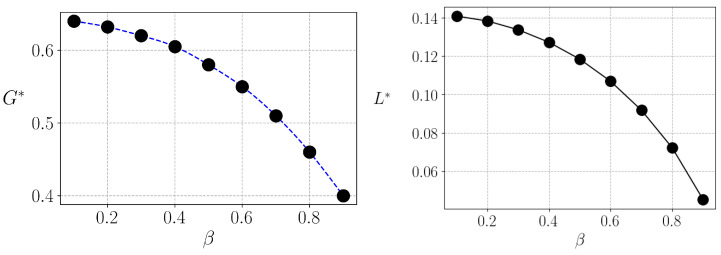
Equilibrium Gini index (**left**) and liquidity (**right**) at $\lambda =\lambda^{*}$ as a function of β. Lines are just guides for the eyes, points are the results of the simulations for different values of β.

## Data Availability

On request to benhur.phys@gmail.com.

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
