# Peer review of "Taxes, Inequality, and Equal Opportunities"

_entropy, 2023, doi:10.3390/e25091346_

Round 1

Reviewer 1 Report

The research carried out relates to an intensely debated issue, regarding how the taxation system can contribute to reducing inequalities and stimulating economic development. The authors may consider the following suggestions:

  - rewriting the abstract, so that the readers can clearly identify the objectives, the hypotheses, the methodology applied and the results obtained;

- completing the introduction with the research hypotheses, so that they can be validated/invalidated after applying the research methodology;

- validation of the results obtained by reporting to other researches;

- completing the conclusions with the limits of the research, future research directions, the clear highlighting of fiscal policy measures that can be adopted by the authorities to reduce inequalities;

- a distinct Discussion section can be included, in which to summarize the results obtained, by reporting to other researchers;

- the review of the entire paper to highlight the novelty of the research, considering the fact that a large part of it presents the same aspects as those analyzed in the paper indicated in reference 31.

Reviewer 2 Report

Overall, this manuscript addresses an important and timely topic regarding the use of taxation and redistribution mechanisms to reduce inequality. The scientific questions and models explored are relevant and could provide useful insights. However, there are areas for improvement that the authors should address to strengthen the manuscript.

Major comments:

The introduction provides a good overview of the inequality problem and existing economic theories. However, it could be expanded to situate the study more clearly within the context of previous relevant research on redistribution and taxation models. Some key works in this area should be cited and discussed to frame the research gap that the current study aims to fill.

The model and method sections could be expanded to provide more details about the simulation parameters, initial conditions, and algorithms used. More thoroughly explaining the model set-up will improve clarity and allow the reader to evaluate the appropriateness and validity of the approaches used.

The results presented in Figures 1 through 5 reveal interesting trends regarding the impact of tax rates and redistribution mechanisms on inequality and economic activity. However, additional analysis and discussion of these results is needed. The authors should interpret the findings within the context of economic theory and previous literature, discuss caveats and limitations, and establish the policy implications of their results.

Many of the claims in the discussion and conclusion sections lack sufficient support from the results presented. The authors need to more rigorously back up their assertions with evidence from their models and analysis. Some of the findings also appear overstated given the limitations of the study, so the authors should moderate their language accordingly.

There are some minor technical errors and issues with the explanations, references, and language that should be addressed to improve the manuscript.

Round 2

Reviewer 1 Report

The authors responded, for the most part, to the suggestions made.

Reviewer 2 Report

Overall this is a significant improvement over version 1. The authors have addressed many of the comments and criticisms raised in the first review. Some key improvements include:

The introduction and literature review sections have been expanded to provide more context and justification for the research questions. There is a clearer delineation between previous findings and the novelty of this work.

The mathematical models and simulations are described in much greater detail, making them transparent and reproducible. Methodological approaches have been strengthened.

Results are presented more clearly and concisely. Figures and illustrations help to explain key findings. More simulations were performed to verify consistency.

The discussion section now ties the findings back to the original research questions more explicitly. Broader implications and limitations are acknowledged.

References have been formatted consistently and appear to be complete based on my spot checking.

Some areas that could still be improved:

The theoretical framework introducing concepts like condensation could be streamlined for clarity.

More explanation of assumptions like uniform distributions would strengthen realism.

Comparisons to other models like the base Yard-Sale model could be expanded.

Quantitative analysis of optimal tax rates versus savings behaviors could be explored further.
